# E2PNet: Event to Point Cloud Registration with Spatio-Temporal Representation Learning

**Xiuhong Lin**[ab], **Changjie Qiu**[ab]\*, **Zhipeng Cai**[c], **Siqi Shen**[ab]†, **Yu Zang**[ab],
**Weiquan Liu**[ab], **Xuesheng Bian**[ae], **Matthias Müller**[d], **Cheng Wang**[ab]

[a]Fujian Key Lab of Sensing and Computing for Smart Cities,
School of Informatics, Xiamen University (XMU), China.
[b]Key Laboratory of Multimedia Trusted Perception and Efficient Computing, XMU, China.
[c]Intel Labs. [d]Apple Inc. [e]Yancheng Institute Of Technology, China.
{siqishen,cwang}@xmu.edu.cn, zhipeng.cai@intel.com
{xhlinxm,qiuchangjie,xsbian}@stu.xmu.edu.cn

## Abstract

Event cameras have emerged as a promising vision sensor in recent years due to their unparalleled temporal resolution and dynamic range. While registration of 2D RGB images to 3D point clouds is a long-standing problem in computer vision, no prior work studies 2D-3D registration for event cameras. To this end, we propose E2PNet, the first learning-based method for event-to-point cloud registration. The core of E2PNet is a novel feature representation network called Event-Points-to-Tensor (EP2T), which encodes event data into a 2D grid-shaped feature tensor. This grid-shaped feature enables matured RGB-based frameworks to be easily used for event-to-point cloud registration, without changing hyper-parameters and the training procedure. EP2T treats the event input as spatio-temporal point clouds. Unlike standard 3D learning architectures that treat all dimensions of point clouds equally, the novel sampling and information aggregation modules in EP2T are designed to handle the inhomogeneity of the spatial and temporal dimensions. Experiments on the MVSEC and VECtor datasets demonstrate the superiority of E2PNet over hand-crafted and other learning-based methods. Compared to RGB-based registration, E2PNet is more robust to extreme illumination or fast motion due to the use of event data. Beyond 2D-3D registration, we also show the potential of EP2T for other vision tasks such as flow estimation, event-to-image reconstruction and object recognition. The source code can be found at: E2PNet.

## 1 Introduction

Computer vision aims to understand and reconstruct the world from 2D observations [2]. Establishing the relationship (pose or correspondences) between 2D data and the 3D world is an important step toward this goal. Conventional 2D-3D registration mostly relies on multi-view geometry [3, 4, 5]. Specifically, 3D (key) points are first triangulated [6, 5] using multiple images with proper disparity. Subsequently, the corresponding 2D features [7, 8, 9] are attached to 3D points for subsequent 2D-3D registration. This framework is by design image-based, which limits the use of other sensors that obtain 3D point clouds directly (e.g., LiDAR, depth cameras).

Recently, learning-based methods for direct 2D-3D registration have been proposed [10, 1, 11, 12]. However, these methods require high-quality RGB or grayscale images which are not always feasible

---

\*Xiuhong Lin and Changjie Qiu contribute equally to this work.
†Corresponding author

37th Conference on Neural Information Processing Systems (NeurIPS 2023).

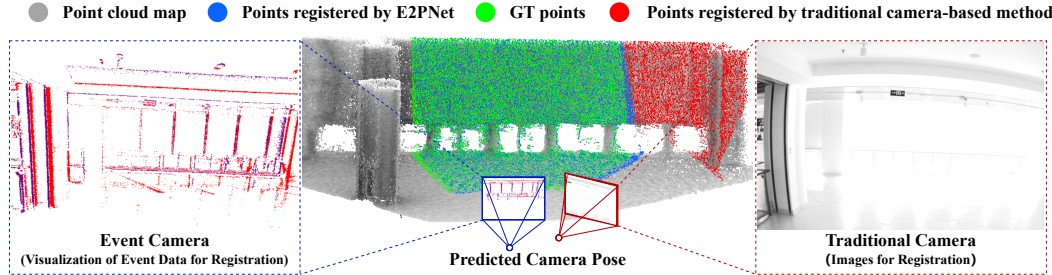

● Point cloud map   ● Points registered by E2PNet   ● GT points   ● Points registered by traditional camera-based method

**Event Camera**
(Visualization of Event Data for Registration)

**Predicted Camera Pose**

**Traditional Camera**
(Images for Registration)

Figure 1: **Teaser.** This work proposes the first learning-based event-to-point cloud registration method E2PNet. As shown in the figure, challenging illumination/fast motion may cause RGB-based methods (e.g., LCD [1]) to fail, resulting in erroneous registration (red camera). Using event data as input, E2PNet is robust to such illumination change and accurately recovers the camera pose (blue).

in practice. Challenging environments with extreme illumination or motion blur exist in many applications, which may cause the algorithm to fail completely (see Fig. 1 for an example).

Unlike traditional cameras, the novel information acquisition principle of event cameras [13] makes them less susceptible to challenging illumination and motion blur. This advantage motivates us to design a robust 2D-3D registration method for event cameras.

High dynamic range (HDR, up to 120 dB [13]) and high temporal resolution bring excellent performance to event cameras but also create new challenges. The unstructured data stream of event cameras is sparse in the spatial plane but dense (up to 300 million events per second [14]) and non-uniform in the temporal dimension. Existing methods mostly process events as image-like feature maps [15, 16] or voxels [17, 18, 19], which arguably loses spatio-temporal details. Although event cameras have been used for other vision tasks such as optical flow estimation [15, 18, 20], moving object detection [21, 22] and high dynamic image reconstruction [23, 24], the task of *event-to-point cloud registration (E2P)* has not yet been studied.

There are several challenges. First, event cameras only capture the surface appearance of the scene, whereas 3D point clouds encode geometric structures, which is inherently challenging [11]. Second, event cameras have lower resolution than traditional cameras and only capture the brightness variation rather than the brightness itself. Finally, event data and point clouds can both be considered as unstructured 3D data, which are harder to process than 2D images. Moreover, the lack of an E2P dataset with accurate annotations limits the research of such cross-modal registration methods.

To address these problems, we present E2PNet, the first E2P framework and the corresponding dataset for end-to-end learning. Specifically, we propose a modular event representation network called Event-Points-to-Tensor (EP2T), which applies local feature aggregation with different temporal and spatial distance weights and spatio-temporal separated attention mechanisms to effectively encode sparse and in-homogeneous event data. In the meantime, EP2T adaptively aggregates local neighborhood features with uneven event data in the temporal and spatial dimensions and outputs a *structured* (grid-shaped) feature tensor which can be directly plugged into image-based 2D-3D registration frameworks. E2PNet is capable of E2P registration for indoor scenes of different scales (single room to one floor) and produces the corresponding 6-DOF transformation between the event camera and 3D point clouds. As shown in Fig. 1, replacing the RGB inputs with event data can significantly improve the registration performance on challenging data. Our main *contributions* are:

- We propose EP2T, the first representation learning architecture to transform event spatio-temporal point clouds into a grid-shaped feature tensor. EP2T can adaptively learn how to extract critical spatio-temporal information according to downstream tasks, and can be applied as a plug-and-play module to existing event camera processing networks.

- We propose E2PNet, the first approach that allows direct registration of 2D event cameras and 3D point clouds. To facilitate follow-up research, we also propose a framework to construct E2P datasets using existing SLAM datasets.

- We conduct extensive experiments and analysis to demonstrate the effectiveness of E2PNet for E2P, and the potential of the EP2T module to benefit in other diverse tasks.

## 2 Related Works

### 2.1 Event Representation

The asynchronous unstructured event data poses a great challenge for computer vision. While spiking neural networks (SNN) [25] can directly use the raw data of event cameras, they require special hardware and has low flexibility. A more common approach is to process batches of event data via an intermediate representation, e.g., an image of positive and negative events over a certain horizon. The processing window is usually set to a fixed number of events (FEN) [26, 20] or a fixed interval time (FIT) [15, 20].

**Hand-crafted** representations usually convert raw event data into 2D grid-shaped feature maps with sensor resolution using the statistics of raw data, for example, event counting [27], average/latest timestamp of events on each pixel [15] or temporal distance-weighted statistics [17]. While these strategies can approximate the distribution of event data with respect to space and time, some important details (i.e. spatio-temporal correlation) may be lost due to the aggregation. To address this shortcoming, another set of methods [28, 29, 30, 31] discards most of the events and only keeps the latest few events for each pixel and analyzes them in detail. However, these methods cannot encode distant spatio-temporal relationships. More recent work tries to incoporate these relationships by representing event data as graphs [32, 33, 34, 35]. Although more spatio-temporal information can be extracted, building a graph structure is complicated and computationally expensive. Each hand-crafted representation has different advantages and disadvantages. Hence, choosing the right representation for a given tasks is often not trivial. The proposed E2PNet also utilizes hand-crafted methods [15, 20, 17]. However, we use them in conjunction with our *learned* features instead of raw event data to produce the final grid-shaped feature tensors.

**Learning-based** representations train models to adaptively extract critical information based on the specific task. However, the inhomogeneity and lack of structure make event representation learning challenging. EST [16] and DDES [30] propose to learn tensor-based representations, which regard the events of each pixel as pulse signals and optimize the impulse response function. ECSNet [36] uses 3D point cloud learning methods to extract point-based features. However, these feature points are incompatible with the processing pipeline of downstream tasks, and generalize poorly to other tasks. In contrast, we aim to fully extract the spatio-temporal correlation of the event data and transform it into a grid feature tensor representation for better compatibility.

### 2.2 2D-3D Registration

This work focuses on the direct 2D-3D registration problem, which has been studied for conventional images. Li et al. [37] use the multi-view projection image of 3D objects to extract the corresponding HOG features, and train a feature mapping network to push the features from the same 2D-3D objects closer. 3DTNet [38] uses RGB-D point clouds and images to train distinctive 3D descriptors. However, these methods can only be used to register images to *colored* point clouds, which is not always feasible. E2PNet does not generate features based on color information, hence it can be applied to conventional point clouds without colors. 2D3D-MatchNet [10] and LCD [1] learn consistent feature representations of 2D images and 3D point clouds, realizing 2D-3D registration through cross-modal feature retrieval. P2-Net [12] learns the detection and matching of potential feature points from images and RGB-D points. DeepI2P [11] transforms image-point cloud registration into point cloud visibility classification and then maximizes the number of visible points by optimizing the camera pose. A common drawback of these image-based methods is that they require high-quality images to perform well, which cannot be guaranteed in applications with challenging illumination conditions or camera motion. As shown later in the experiments, introducing event data for 2D-3D registration makes E2PNet much more robust in challenging environments.

## 3 Methodology

E2PNet treats the input event data as spatio-temporal point clouds. Before explaining how this can be done, we briefly introduce the generation process of event data here. During data capture, each pixel of the event camera responds independently and asynchronously to illumination changes. Specifically, given an image of size $(H, W)$, denote $I(t_i, h, w)$ as the illumination at time $t_i$ for the pixel at $(h, w)$.

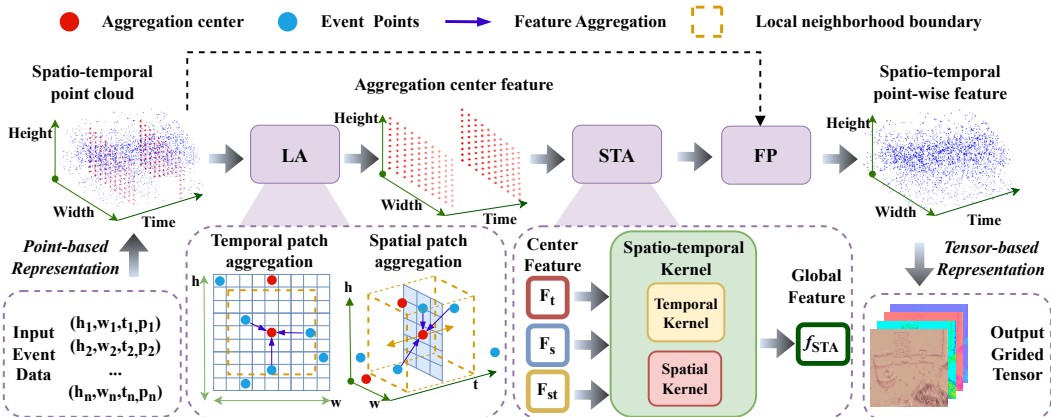

Figure 2: **The proposed Event-Points-to-Tensor (EP2T) network.** EP2T treats event data as spatio-temporal point clouds. Local Aggregation (LA) first generates feature aggregation centers and aggregates local features from neighboring points. Subsequently, Spatio-temporal Separated Attention (STA) extracts global features using attention. To handle the spatio-temporal inhomogeneity problem, LA and STA separate spatial and temporal information during computation. Next, Feature Propagation (FP) propagates features from aggregation centers to each event point based on the distance. Finally, grid-shaped feature tensors are obtained by different yet complementary event tensorization methods.

When the inequality

$$\|\log(I(t_i, h, w)) - \log(I(t_{i-1}, h, w))\| > \tau \tag{1}$$

is satisfied. This pixel will generate an impulse response called an *event*, which is expressed as $\mathbf{e} = (h, w, t_i, p)$. The polarity $p \in \{+1, -1\}$ indicates the direction of the brightness change (increasing or decreasing) and $\tau$ is the trigger threshold. $t_{i-1}$ represents the time when the event was triggered last time at pixel $(h, w)$. The imaging plane and the time axis can be combined to establish a three-dimensional spatio-temporal coordinate system, where events characterized by coordinates $(h, w, t_i)$ collectively form a spatio-temporal point cloud.

In order to compute the pose of the camera w.r.t. a 3D point cloud, E2PNet first encodes the spatio-temporal point clouds into a 2D grid-shaped feature tensor. This process relies on EP2T, which is the core component of E2PNet and will be introduced in detail in Sec. 3.1. After obtaining the encoded feature tensor, we can formulate the event-to-point registration (E2P) problem as the registration of 2D feature maps and 3D point clouds, which can be handled by mature image-based frameworks [1, 11]. Since EP2T is differentiable, the training of E2PNet is simply done by connecting EP2T with the registration network and performing back-propagation in the same way as the original registration network. Empirically we found that directly adding EP2T to the registration network without changing any hyper-parameter works reasonably well, which is another advantage of E2PNet.

### 3.1 Spatio-Temporal Reprsentation Learning with Event-Points-to-Tensor (EP2T)

As illustrated in Fig. 2, EP2T consists of three consecutive modules: Local Aggregation (LA), Spatio-temporal Separated Attention (STA), and Feature Propagation (FP).

Given a spatio-temporal point cloud derived from the input event data, LA evenly samples a set of feature aggregation centers in the spatial and temporal domains. For each sampled center, LA computes the locally aggregated features using the neighboring input points. To produce the global feature, STA performs attention over these aggregated features, with a special design to handle the spatio-temporal inhomogeneity. To produce features on each input point (from aggregation centers), FP propagates each feature from STA to the neighboring points based on the distance. Finally, we employ classical tensorization methods to process the point-wise features into grid-shaped tensors.

The novelty of LA (see Sec. 3.1.1 for more details) and STA (Sec. 3.1.2) lies in the separation of spatial and temporal domains, which is designed to handle the inhomogeneity of spatio-temporal point clouds. FP (Sec. 3.1.3) propagates aggregated features to neighboring spatio-temporal points so

that standard point-wise tensorization can be used to generate grid-shaped outputs, which enables the use of RGB-based 2D-3D registration frameworks for E2P.

### 3.1.1 Local Aggregation (LA)

Given an event camera of resolution (H, W), we extract N events from the input and convert them into a spatio-temporal point cloud with N points. Unlike conventional point clouds, the spatial and temporal domains of our inputs naturally have different metric scales, which makes existing point cloud representation networks [39] ineffective for feature encoding (see Sec. 4.2).

To address this problem, the proposed LA module aggregates features across the spatial and temporal domains seperately. Specifically, for each input point $\mathbf{e} = (h, w, t, p)$ with $p \in \{-1, 1\}$ representing the brightness change direction, we first normalize it via $\hat{\mathbf{e}} = (\frac{h}{H}, \frac{w}{W}, \frac{t}{t_{\max}-t_{\min}}, p)$, where $t_{\max}$ and $t_{\min}$ are the maximum and minimum time stamps of the input point cloud.

The feature aggregation process of LA is inspired by PointNet++ [39]. There are two steps during aggregation: (1) generate a set of aggregation centers; (2) calculate the aggregated feature for each center by employing neighboring points.

PointNet++ [39] generates aggregation centers using Farthest Point Sampling (FPS) [40], which is relatively slow in event-based applications that require a high response speed. Furthermore, FPS is not suitable for spatio-temporal point clouds with non-uniform density distributions (might generate redundant points in dense regions and fewer points in sparse regions). To address this issue, LA uses uniform sampling with a fixed spatial and temporal interval to generate uniformly distributed aggregation centers. Specifically, given the normalized point cloud $\hat{E}$ spanning over a spatio-temporal cube of size $(H, W, T)$, we generate a set of aggregation centers $C$ by sampling $M$ (30*30*5 in our experiments) uniform grid locations within this cube.

To aggregate features from neighboring points, PointNet++ [39] first selects $K$ nearest neighbors from $\hat{E}$ for each $\mathbf{c}_j \in C$ and then replaces neighbors whose Euclidean distance is too far away from $\mathbf{c}_j$ with the point closest to $\mathbf{c}_j$. After selecting neighboring points, a multi-layer convolution is performed to produce the aggregated feature. Using Euclidean distance omits the inhomogeneity of the spatial and temporal domains, which exists even with our initial normalization. Hence, we propose to separate the spatial and temporal domains during neighborhood selection. Specifically, we change the distance metric during the selection of nearest neighbors ($K = 64$ in LA) to $\alpha d_{\text{Space}}(\hat{\mathbf{e}}_i, \mathbf{c}_j) + \beta d_{\text{Time}}(\hat{\mathbf{e}}_i, \mathbf{c}_j)$. We also use two distances *separately* in the spatial and temporal domains to choose points for replacements, i.e., we replace a neighboring point $\hat{\mathbf{e}}_i$ with the closest point to $\mathbf{c}_j$ if $d_{\text{Space}}(\hat{\mathbf{e}}_i, \mathbf{c}_j) \geq \alpha$ or $d_{\text{Time}}(\hat{\mathbf{e}}_i, \mathbf{c}_j) \geq \beta$. $d(\cdot)$ is the mean squared distance of the corresponding domain.

Separating the spatial and temporal domains during feature aggregation addresses the inhomogeneity problem from two aspects. (1) We can provide constraints to the spatial and temporal domains separately to prevent choosing neighboring points with a small distance in one domain but a far distance in another. (2) We can provide different $\alpha$'s and $\beta$'s to create multiple aggregated features that focuses on different domains. For the second aspect, we produce three sets of features in LA by setting $\alpha = \{0.8, 0.1, 0.5\}$ and $\beta = \{0.1, 0.8, 0.5\}$ respectively, which allows the aggregated feature to focus more on spatial, temporal, and spatio-temporal domains.

### 3.1.2 Spatio-temporal Separated Attention (STA)

After generating the feature centers and the locally aggregated features, we use the STA module to extract global features and merge features from different domains into the uniform spatio-temporal domain. Similar to LA, STA also separates the spatial and temporal domains to effectively handle the inhomogeneity problem.

$\mathbf{F}_t$, $\mathbf{F}_s$, and $\mathbf{F}_{st}$ in Fig. 3 denote the aggregated temporal, spatial and spatio-temporal features from LA respectively. We first create global features for the spatial and temporal domains separately using the temporal and spatial residual kernels. Each of these two kernels performs self-attention over features from the same domain, and cross-attention between features from different domains. Subsequently, we merge them into a temporal and spatial residual using retention gates (sigmoid and tanh functions) inspired by [41, 42]. After producing the residuals for spatial and temporal domains separately, we merge them into the spatio-temporal domain using the spatio-temporal aggregated feature $\mathbf{F}_{st}$ to produce the global spatio-temporal feature $f_{\text{STA}}(\mathbf{c}_j)$ for each aggregation center $\mathbf{c}_j$.

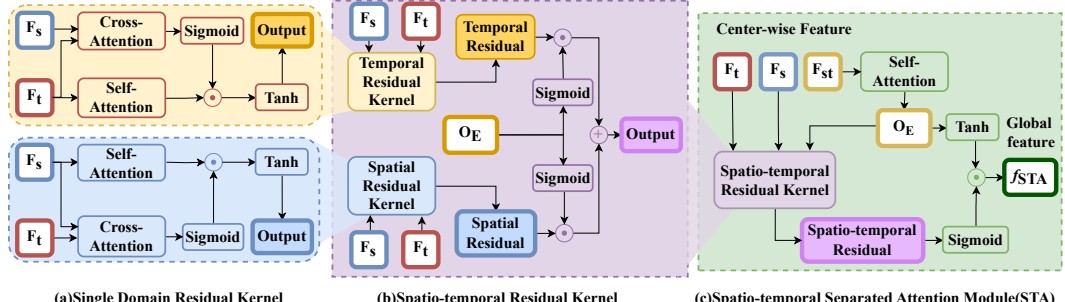

Figure 3: **Detailed structure of the proposed STA module.** (a) Encode features for the spatial and temporal domains separately using the spatial and temporal residual kernels. (b) Merge both residuals into the spatio-temporal domain. (c) Apply the spatio-temporal residuals to the spatio-temporal features.

### 3.1.3 Feature Propagation (FP) and Tensorization

After producing the global features at the aggregation centers, we propagate them to neighboring input event points using a weighted average to get a feature for each input point. Specifically, given the global features $F_{\text{STA}} = \{f_{\text{STA}}(\mathbf{c}_j)\}$ and the normalized event points $\hat{E} = \{\hat{\mathbf{e}}_i\}$, we compute the propagated feature by

$$f_{\text{FP}}(\hat{\mathbf{e}}_i) = \frac{\sum_j w_{i,j} f_{\text{STA}}(\mathbf{c}_j)}{\sum_j w_{i,j}}, \tag{2}$$

where $w_{i,j} = \frac{1}{\max(||\hat{\mathbf{e}}_i - \mathbf{c}_j||_2^2, 10^{-5})}$. $10^{-5}$ is used to prevent division by zero.

Finally, to convert point-based features into grid-shaped features, we use three different yet complementary event tensorization methods [15, 20, 17] to convert the set of $f_{\text{FP}}(\hat{\mathbf{e}}_i)$ into three different types of 2D grid-shaped sparse feature tensors, and concatenate them to produce the final tensorized feature map.

### 3.2 E2P Dataset Construction

Since there is no dataset for E2P, we use multi-sensor SLAM datasets to construct the ground-truth (GT) labels for training. We select the widely used MVSEC [43] and VECtor [44] to build the MVSEC-E2P and VECtor-E2P datasets, which incorporate LiDAR, traditional cameras and event cameras at the same time. Furthermore, they have an accurate multi-sensor time synchronization and calibration. Through the GT pose provided in these two datasets, we can establish the GT matching relationship between events (images) and point clouds for E2P tasks. A naive way to obtain such GT would be to project point clouds into the image plane, which is slow in practice. Instead, we construct a camera model in the world coordinate system and use this model to construct a quadrangular viewing frustum. Selecting the points in the quadrangular viewing frustum is practically $> 2x$ faster than the naive solution.

## 4 Experiments

In this section, we first demonstrate the effectiveness of the proposed E2PNet for event-to-point cloud registration (Sec. 4.1), which is our primary goal. Then, we conduct ablation studies to verify the effectiveness of individual modules in EP2T (Sec. 4.2). Finally, we analyze the generalization of EP2T on other event-based vision tasks (Sec. 4.3).

### 4.1 Event to Point Cloud Registration

**Datasets and Preprocessing.** We use two representative datasets described in Sec. 3.2. MVSEC [43] uses a 16-beam LiDAR and an event camera with a resolution of (346,260); the event camera can simultaneously generate grayscale images. Since only indoor sequences have high-precision pose

Table 1: **Quantitative results of 2D-3D registration.** Since the three hand-crafted event tensorization representation methods [15, 17, 20] are complementary, we use them jointly to achieve the best performance and call this hybrid method E-Statistic. To fairly compare against point-based representations, we also remove the tensorization process of E2PNet, i.e., w/o TR. The best performing method is highlighted.

| Input Representation | Method | DeepI2P [11](Registration-based) | | | | LCD [1] (Retrieval-based) | | | |
| | | MVSEC-E2P | | VECtor-E2P | | MVSEC-E2P | | VECtor-E2P | |
| | | RE(°)(↓) | TE(m)(↓) | RE(°)(↓) | TE(m)(↓) | RE(°)(↓) | TE(m)(↓) | RE(°)(↓) | TE(m)(↓) |
| Traditional Image | Grayscale Image | 7.922 | 0.370 | 11.343 | 3.176 | 6.335 | 1.347 | 17.879 | 13.200 |
| Event(Tensor-based) | E-Statistic | 6.748 | 0.250 | 10.654 | 3.524 | 4.968 | 1.297 | 11.034 | 9.416 |
| | Tore [19] | 7.465 | 0.192 | 10.542 | 4.565 | 4.855 | 1.350 | 9.521 | 7.254 |
| | Ours | **5.127** | **0.164** | **8.778** | **2.454** | **3.606** | 0.821 | **8.672** | 7.403 |
| Event(Point-based) | ECSNet [36] | 8.075 | 1.612 | 10.149 | 2.542 | 4.985 | 1.263 | 20.740 | 13.284 |
| | Ours(w/o TR) | 6.021 | 0.721 | 8.795 | 3.212 | 4.422 | **0.768** | 9.120 | **6.551** |

ground truth (GT), we use the indoor-$x$ and indoor-$y$ sequences for training and testing respectively, where $x \in [1,3]$ and $y = 4$. VECtor [44] uses a 128-beam LiDAR and an event camera with a resolution of (640,480). Different from MVSEC, VECtor contains various large-scale indoor environments with long trajectories. The GT pose is derived via ICP[45] between LiDAR scans and dense environment point clouds captured by a high-precision laser scanner. We use the units-dolly, units-scooter, corridors-dolly and corridors-walk sequences for training, and the school-dolly and school-scooter sequences for evaluation. For both datasets, we randomly acquire event data in an (256,256) area from event camera data as input; simultaneously we also acquire grayscale images with the same field of view and resolution for comparison.

**Evaluation.** Following the previous standard [46], we use the translation error TE $= ||\mathbf{T}_{GT} - \mathbf{T}_{pred}||_2$ and the rotation error: RE $= \arccos \left( \frac{tr(\mathbf{R}_{GT}^{-1}\mathbf{R}_{pred})-1}{2} \right)$ to evaluate the accuracy of predicted camera poses in E2P. $\mathbf{T}$ and $\mathbf{R}$ represents the translation vector and rotation matrix respectively, and $tr(\cdot)$ is the trace of a matrix.

**Implementation Details.** We follow the FEN [26, 20] principle and acquire 20000 consecutive events at a time and sample $N = 8192$ events from them. Since EP2T has encoded the spatio-temporal information as 2D grid tensors, we can regard E2P as the registration of 2D feature maps with 3D point clouds. We choose LCD [1] and DeepI2P [11] as the representatives to test the performance of E2P using feature retrieval-based and point visibility-based registration. LCD maps the corresponding 2D tensor and 3D point cloud into a high-dimensional feature space, where training aims to make the feature distance as close as possible. During training, we fetch events (image) in the current pose and their corresponding point cloud patch according to the GT pose (see Sec. 3.2). The principle of DeepI2P is to judge whether the input point cloud appears in the corresponding field of view of events (image). During training, we use the visible point clouds for each event view and its surrounding point clouds (enlarged by 30%) for supervision. All methods are implemented using Pytorch. Training is done following the setup of individual baselines on a 3090Ti GPU.

We compare E2PNet against conventional camera-based registration (i.e., the original version of LCD and DeepI2P) and other event representation methods, including tensor-based [19, 15, 17, 20] and point-based [36] methods. To compare against point-based methods, we do not perform the final tensorization (i.e., w/o TR) so that the encoded features are also point-based. Since there is no standard technique to apply point-based event representations to image-based registration frameworks, for fair comparisons, we implement a simple encoder with max-pooling and a linear layer (see appendix for details) to project the point-based features to a global feature vector. Subsequently, we use this vector to replace the original global feature (computed by performing convolution on the tensor-based representation) in both LCD and DeepI2P. For LCD-based methods (retrieval-based), we use the trained model to encode the feature on the target point cloud under different poses to build a point cloud feature library. During inference, we extract the corresponding pose in the feature library

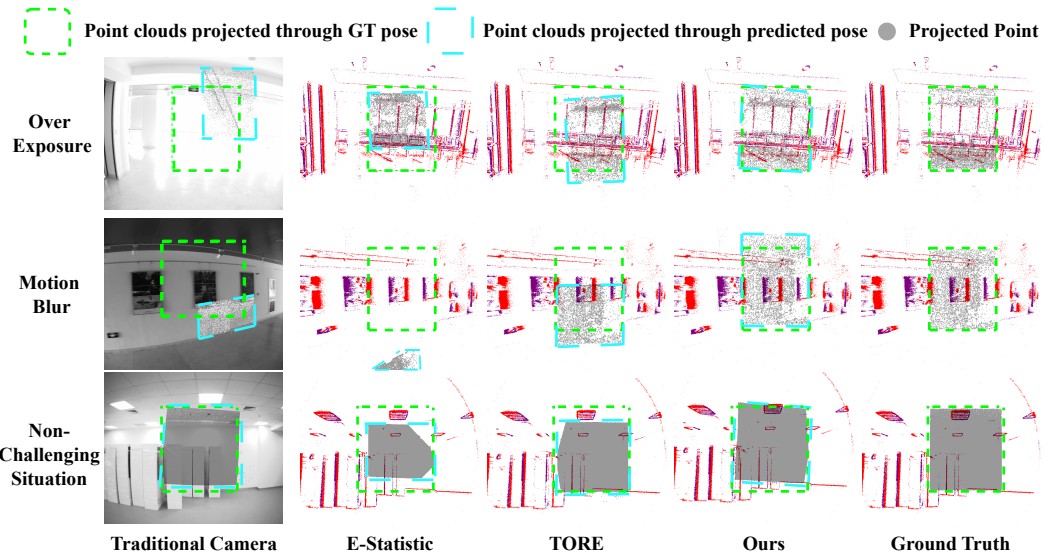

Figure 4: **Qualitative comparison of E2PNet with other methods.** The green and cyan boxes represent the projected position of the 3D point clouds onto the view plane of the camera after rigidly transforming to the camera coordinate system using ground truth pose and predicted pose respectively. Compared to the image-based baseline (LCD), event-based methods are more robust to challenging illumination and motion. E2PNet (ours) performs best among all event-based methods.

Table 2: **Ablation**. Removing individual modules from E2PNet hurt the registration accuracy.

| Method | Ours | Ours w/o S-T separated in STA | Ours w/o STA | Ours w/o LA+FP | Ours w/o LA+STA+FP |
|---|---|---|---|---|---|
| RE(°)(↓) | **3.606** | 3.710 | 3.922 | 4.878 | 4.968 |
| TE(m)(↓) | **0.821** | 0.824 | 0.913 | 1.232 | 1.297 |

that is closest to the feature of the event data. For DeepI2P-based methods (point cloud visibility classification-based), we use the same optimization method as the original method [11].

As shown in Tab. 1, replacing conventional image inputs with event data improves the registration accuracy significantly and consistently *across datasets and baseline methods*. E2PNet performs the best among both tensor-based and point-based E2P methods, outperforming both hand-crafted (E-Statistic) and learning-based methods. Qualitative results in Fig. 4 further show the robustness of E2PNet in different environments. Finally, due to better compatibility, tensor-based methods perform better than point-based methods, and sometimes by a large margin. Note that although point-based representations can be applied to 2D-3D registration, since only a global feature vector of the input (event) image is required, they may not be compatible with tasks that require pixel-level feature maps.

## 4.2 Ablation studies

In this section, we analyze the effectiveness of individual components of E2PNet, i.e., the Local Aggregation module (LA), Spatio-temporal Separated Attention module (STA), Feature Propagation module (FP), and the mechanism for seperating the spatial and temporal domains in STA. We use LCD [1] as our baseline model and conduct experiments on the MVSEC-E2P dataset. As shown in Tab. 2, the registration accuracy reduces as components are removed, showing the importance of each component. Note that since LA and FP are coupled modules (we need FP to propagate features from aggregation centers to individual event points), we add and remove them together.

Table 3: **Results of EP2T on other vision tasks.** "Ours" means the result obtained after incorporating our EP2T module. For a fair comparison, The relevant training settings are the same as the original framework. The dataset used for evaluation is indicated after "@".

| Flow Estimation | | | Event-to-Image Reconstruction | | | | Object Recognition | |
|---|---|---|---|---|---|---|---|---|
| Method | AEE(↓) | Outlier(%)(↓) | Method | MSE(↓) | SSIM(↑) | LPIPS(↓) | Method | Accuracy(↑) |
| cGAN[20]@[43] | 0.34 | 0.06 | cGAN[20]@[43] | 0.479 | 0.431 | 0.450 | EST[16]@[31] | 0.746 |
| cGAN(ours)@[43] | **0.30** | **0.005** | cGAN(ours)@[43] | **0.476** | **0.460** | **0.435** | EST(ours)@[31] | **0.771** |
| EV-Flow[15]@[43] | 0.478 | **0.03** | BTEB[47]@[48] | 0.20 | 0.32 | 0.46 | EST[16]@[49] | 0.949 |
| EV-Flow(ours)@[43] | **0.462** | **0.03** | BTEB(ours)@[48] | **0.16** | **0.32** | **0.42** | EST(ours)@[49] | **0.957** |

## 4.3 Generalization Experiments

Though we focus on the task of E2P in this work, the architecture design of our EP2T module is also applicable to other vision tasks. In this section, we apply EP2T to three different tasks (see below) to show its potential. Specifically, we embed the EP2T module into existing baseline models as an additional information channel and compare the performance. For all baseline models, we use the released source code and the corresponding training setup during experiments.

**Flow estimation** aims to infer the instantaneous speed and direction of the corresponding pixel motion of the object on the imaging plane. Compared to the traditional frame-difference-based optical flow estimation methods, event camera optical flow estimation has a higher temporal resolution, enabling more accurate estimation of object motion in complex scenarios involving fast motion and high rotation rates. We employ the standard networks cGAN [20] and EV-FlowNet [15] as baselines in this experiment. Following standard practice, we utilize the average endpoint error (AEE) [50] and the outlier rate (the ratio of errors greater than three pixels) to evaluate different methods. We use the MVSEC dataset to evaluate these different methods.

**Event-to-image reconstruction** is another canonical task for event cameras in which a high-contrast image with sharp edges is reconstructed from events, making full use of the advantages of the event camera's high contrast and fast response speed. In this experiment, we use BTEB [47] on the ECD dataset [51] and cGAN [20] on the MVSEC dataset [43] as the baselines and use the mean squared error (MSE), structural similarity (SSIM) [52], perceptual similarity (LPIPS) [53] for evaluation.

**Object recognition** with conventional cameras remains challenging due to high latency and motion blur. Event cameras are naturally less sensitive to such problems. In this experiment, we employ the widely used EST [54] network as the baseline. To assess the performance of our proposed methodology, we utilize the publicly available N-caltech101 dataset [49] and N-cars dataset [31] and use the recognition accuracy as the evaluation metric.

As shown in Tab. 3, adding the EP2T module improved the performance of baselines on various datasets and tasks. This result clearly demonstrates the potential of EP2T for other vision tasks. Since our initial motivation to design EP2T is driven by the 2D-3D registration task, we believe the current architecture design and the way to apply our method to other tasks both have further space to be optimized, which we leave as the future works.

## 4.4 Time and Memory Efficiency

We analyzed the time and memory efficiency of the event representation learning module EP2T, as well as the complete E2PNet consisting of EP2T and 2D-3D registration network. As shown in Tab. 4, our E2PNet is slightly slower and uses more memory than baseline methods. However, the overhead introduced by EP2T is arguably small given the significantly improved accuracy.

We also analyzed of the impact of two hyperparameters (number of FEN and number of sampling points) from three perspectives: accuracy, time efficiency, and memory cost in Tab. 5. Selecting an appropriate number of events per batch (FEN) optimizes algorithm performance. A small FEN yields insufficient information, whereas an excessive FEN extends sampling time and limits the exploration of spatiotemporal details. Raising the number of EP2T sampling points enhances spatiotemporal detail capture but escalates computational costs. The major contributor to the higher memory usage is the EP2T network designed based on PointNet++ [39], where the point-wise distance calculation

Table 4: **Time and memory efficiency of different methods.** We compared the event representation module and the Event-to-point cloud registration framework, which consists of the event representation module and the 2D-3D registration network. All experiments were performed with a batch size of 1 on a 3090Ti GPU using the MVSEC-E2P dataset. We jointly use three hand-crafted event tensorization representation methods [15, 17, 20] to achieve the best performance and call this hybrid method E-Statistic. Compared to the baselines, E2PNet demonstrated significantly higher accuracy and marginally lower but still acceptable time and memory efficiency.

| Method | 2D-3D registration using LCD [1] | | | | Event representation | |
|---|---|---|---|---|---|---|
| | Grayscale Image | E2PNet | E-statistic | ECSNet [36] | EP2T | E-statistic |
| Time(ms)($\downarrow$) | 55.29 | 110.62 | **55.00** | 81.82 | 42.8 | **3.16** |
| Space(MB)($\downarrow$) | 2218 | 7890 | **2216** | 2530 | 4650 | **0.68** |
| RE(°)($\downarrow$) | 6.336 | **3.606** | 4.968 | 4.985 | × | × |
| TE(m)($\downarrow$) | 1.347 | **0.821** | 1.297 | 1.263 | × | × |

Table 5: **The influence of varying FEN and the number of EP2T sampling points on the accuracy and efficiency of E2PNet.** FEN represents the event count in each batch, from which E-statistic is employed to extract global spatial features. Our EP2T algorithm selects some sampling points from the events processed in each batch as the aggregation center of the spatiotemporal information. All experiments are conducted on MVSEC-E2P dataset using LCD [1] framework, with inference performed on a 3090ti GPU and a batch size of 1.

| Number of FEN | EP2T sampling points | RE(°)($\downarrow$) | TE(m)($\downarrow$) | Time(ms)($\downarrow$) | Space(MB)($\downarrow$) |
|---|---|---|---|---|---|
| 20000 (Ours) | 512 | 3.813 | 1.089 | **65.51** | **2992** |
| | 4096 | 3.777 | 0.954 | 79.55 | 4394 |
| | 8192 (Ours) | 3.606 | 0.821 | 110.62 | 7890 |
| | 16384 | **3.247** | **0.789** | 144.09 | 12190 |
| 8192 | 8192 | 3.479 | 1.074 | 90.34 | 7258 |
| 60000 | | 4.242 | 1.426 | 109.50 | 7908 |

operation leads to memory growth that scales with the number of EP2T sampling points. However, when reducing the number of EP2T sampling points to 512, E2PNet runs at a similar speed (65.5ms) and memory cost (2.99GB) as other baselines yet still has higher accuracy.

In summary, our EP2T module is both effective and adaptable, introducing a novel approach for extracting spatiotemporal correlations and detailed information from event camera data. According to different performance requirements and available computing resources, the number of EP2T sampling points and sampling strategies can be flexibly adjusted to strike an efficiency-accuracy trade-off. The impact of different sampling strategies can also be viewed in the Appendix.

## 5   Conclusion and Limitation

In this work, we propose E2PNet, the first learning-based event-to-point cloud registration (E2P) method. We design a novel Event-Points-to-Tensor (EP2T) network, which encodes event data into feature tensors in the shape of a 2D grid such that mature RGB-based 2D-3D registration frameworks can be used for the E2P task. EP2T treats event data as spatio-temporal point clouds and separates the temporal and spatial domains for processing to handle the inhomogeneity of the event point cloud. Our experiments show that E2PNet is superior compared to other learning-based or handcrafted event representation methods and the traditional RGB-based registration in extreme environments (overexposure, motion blur). In addition, we also show the potential of the EP2T module in vision tasks beyond 2D-3D registration. Since E2PNet is inspired by 3D point-based learning architectures [39], similar limitations exist. For instance, an excessive number of sampling points will result in large memory consumption and efficiency degradation of E2PNet. However, further efficiency improvements are possible by some optimization techniques like delayed aggregation [55], sparse convolution [56]. We believe that solving this problem is an interesting and important future research direction.

# 6 Acknowledgement

This work was supported by the National Natural Science Foundation of China (61971363, 62301473); by the FuXiaQuan National Independent Innovation Demonstration Zone Collaborative Innovation Platform (No.3502ZCQXT2021003); by the Fundamental Research Funds for the Central Universities (No. 20720230033); by PDL (2022-PDL-12); by the China Postdoctoral Science Foundation (No.2021M690094). We would like thank the anonymous reviewers for their valuable comments.

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
