# Appendix
# E2PNet: Event to Point Cloud Registration with Spatio-Temporal Representation Learning

## A  Appendix

In this appendix, we first provide the implementation details of our proposed E2PNet and corresponding datasets preprocessing (see Sec. A.1). Subsequentially, the details of evaluation metrics used for evaluating the generalization of the EP2T module are presented (see Sec. A.2). Finally, we analyze the changes in registration precision under different network settings (see Sec. A.3)and report additional quantitative on other vision-based methods (Event-to-Image Reconstruction, Flow Estimation).

### A.1  Implementation Details

In this section, we first present the implementation details of our proposed Event-Points-to-Tensor (EP2T) module (see Sec. A.1.1). Based on EP2T, we introduce the training strategy of E2PNet and the detailed construction method of event-to-point cloud registration (E2P) dataset (see Sec. A.1.2 and Sec. A.1.3). Finally, we introduce the dataset used for generalization experiments (see Sec. A.1.4).

### A.1.1  Architecture of EP2T

**LA and STA.**   In the Local Aggregation module (LA), after selecting the neighborhood of each aggregation center, we calculate the spatio-temporal distance between each center and the corresponding neighborhood points. For each neighborhood, we use a block consisting of three 2D convolutional with kernel size $= 1 * 1$ and batch normalization layers to extract local aggregation distance information. The output dimensions of each layer in the block are (16, 32, 64), and a ReLU activation function is added after the last normalization layer.

Subsequently, the features of the aggregation centers will be sent to the Spatio-temporal Separated Attention module (STA). In STA, we use two different attention mechanisms to jointly extract global features [1]. Specifically, we use three different 1D convolution with output channel $= 64$ and kernel size $= 1$ to obtain the query matrix, key matrix, and value matrix. To prevent paying too much attention to a certain dimension, we use the multi-head attention ($Head = 4$) mechanism to learn different attention patterns. The difference between self-attention and cross-attention is whether to use another domain to calculate the key matrix.

**Tensorized Representation.**   After the LA, STA and FP (Feature Propagation) modules, each event point is embedded with high-dimensional (64 channels) spatio-temporal features $f_{\text{FP}}(\hat{\mathbf{e}}_i)$ . To convert point-based features into grid-shaped features, we pioneered the combination of 3D point cloud learning-based and hand-crafted-based methods. Specifically, we separately superimpose the features of each channel to obtain a gridded feature tensor consisting of 64 channels. After that, we perform

channel max pooling on all feature points to obtain the global features $f_{\mathrm{G}}(\hat{\mathbf{e}}_i)$ of each point. Finally we first use three different yet complementary event tensorization methods [2, 3, 4] to convert the set of $f_{\mathrm{G}}(\hat{\mathbf{e}}_i)$ into different types of 2D grid-shaped sparse feature tensors, and concatenate them to produce the final tensorized feature map. Specific tensorization methods are as follows:

***Sum of Features*** [2] separately superimpose the features of each channel to obtain a gridded feature tensor consisting of 64 channel. The feature value on each polarity and channel can be obtained by formula (1)

$$\boldsymbol{F_{SF}}_{(h,w,c)} = \sum_{i=1}^{N} p_i * k_b\left(h - h_i\right) * k_b\left(w - w_i\right) * f_{\mathrm{FP}}^{c}(\hat{\mathbf{e}}_i) \tag{1}$$

where polarity $p \in \{-1, 1\}$, channel $c \in [1, 64]$, events $\mathbf{e_i} = (h_i, w_i, t_i, p_i)$ and $k_b(x) = \max(0, 1 - |x|)$ is the bilinear sampling kernel, here we use it as an indicator function. Finally, we normalize them to get a set of grid-shaped feature tensors with shape $64 * H * W$.

***Event counting*** [2] separates the positive and negative ($p_i$ represents the polarity) events and handles them separately. The feature value on each polarity channel can be obtained by formula (2)

$$\boldsymbol{F_{EC}}_{(h,w,p)} = \sum_{i=1}^{N} k_b\left(h - h_i\right) * k_b\left(w - w_i\right) * k_b\left(p - p_i\right) * f_{\mathrm{G}}(\hat{\mathbf{e}}_i) \tag{2}$$

where $p \in \{-1, 1\}$, events $\mathbf{e_i} = (h_i, w_i, t_i, p_i)$ and $k_b(x) = \max(0, 1 - |x|)$ is the bilinear sampling kernel, here we use it as an indicator function. $\boldsymbol{F_{EC}}$ superimposes all the features on the same pixel position. Finally, we normalize them to get a set of grid-shaped feature tensors with shape $2 * H * W$.

***Event stacking*** [3] divides event clouds into *B* blocks (*B*=3 in EP2T) equally in time dimension. Given a set of *N* input events $\{e_i\}_{i \in [1,N]}$, the events are arranged in the order of appearance time $t_i$, i.e. in ascending order of the value of $t_i$. *N* events are re-divided into *B* blocks according to time $t_i$, the time range of each event stream is $\left[\frac{(b-1)*\Delta t}{B}, \frac{b*\Delta t}{B}\right]$, where $b \in [1, B]$ and $\Delta t = t_N - t_1$. The calculation method of each blocks is as follow:

$$\boldsymbol{F_{ES}}_{(h,w,b)} = \sum_{i=1}^{N} p_i * k_b\left(h - h_i\right) * k_b\left(w - w_i\right) * f_{\mathrm{G}}(\hat{\mathbf{e}}_i)) \tag{3}$$

where $k_b(x) = \max(0, 1 - |x|)$. Finally, we normalize $\boldsymbol{F_{ES}}$ to get a set of grid-shaped feature tensors with shape $3 * H * W$. Our modification to the method described in the paper [3] involves multiplying the event eigenvalues by their corresponding positive and negative polarities, and then accumulating them, rather than determining the overall eigenvalue sign solely based on the polarity of the latest event. Positive and negative events can cancel each other out, which aligns with the concept of event polarity as defined by brightness change direction.

***Event spatio-temporal Voxelization*** [4] not only divides the event point clouds into *B* blocks (B=3 in EP2T) according to time but also takes the time distance between each event points and the sampling points as one of the weights of feature aggregation.

$$\boldsymbol{F_{EV}}_{(h,w,b,p)} = \sum_{i=1}^{N} k_b\left(h - h_i\right) * k_b\left(w - w_i\right) * k_b\left(b - b_i^*\right) * k_b\left(p - p_i\right) * f_{\mathrm{G}}(\hat{\mathbf{e}}_i)) \tag{4}$$

where $b \in [1, 3]$, $p \in \{-1, 1\}$, $b_i^* = \frac{B*(t_i - t_1)}{t_n - t_i}$ and $k_b(x) = \max(0, 1 - |x|)$. Finally, we normalize $\boldsymbol{F_{EV}}$ to get a set of grid-shaped feature tensors with shape $2 * 3 * H * W$. This method not only encodes the spatial distribution information of events, but also contains the order in which the events are triggered, giving more granular weights to the spatio-temporal features embedding clouds. In general, combining the above three event tensorization representation methods, we jointly construct a sparse tensor with multiple horizons through the coordinate and embedding feature information of the event clouds.

### A.1.2 Training strategy of E2PNet

Our proposed E2PNet is trained with two strategies (tensor-based and point-based). The tensor-based training strategy is straightforward. After we use EP2T to encode event data into a grid-shaped tensor, we directly modify the number of input channels (75 in event-based method, 3 for the original RGB-based method) in the baseline to fit the feature channels of EP2T. Specifically, in DeepI2P [5], we replace the number of input channels of the first layer in the backbone (resnet34) in the image branch. In LCD [6], we replace the first convolutnatbibion layer of the image branch in the encoder part. As for the point-based strategy, we add a channel max-pooling layer across the feature dimension to obtain global features equivalent to the number of channels. To ensure the number of input channels in subsequent networks is the same as tensor-based methods, we add a linear layer with output channel = 64 after channel max-pooling. Finally, in both image brach, we replace the whole encoder backbone of DeepI2P and the encoder part of LCD with this global feature. Other settings remain unchanged in the original paper [5, 6].

### A.1.3 Datasets Preprocessing for E2PNet

Since there is no dataset for the E2P task, we propose to use multi-sensor SLAM datasets to construct the registration relationship GT between event cameras and 3D point clouds. We select MVSEC [7] and VECtor [8] as these datasets use LiDAR, traditional cameras and event cameras simultaneously, and have good calibration. With these calibration parameters and pose GT, we can establish the GT matching relationship between events (images) and point clouds for E2P tasks through the camera projection model. (see Eq. 5).

$$\mathbf{Z}\begin{pmatrix} u \\ v \\ 1 \end{pmatrix} = \begin{pmatrix} f_x & 0 & c_x \\ 0 & f_y & c_y \\ 0 & 0 & 1 \end{pmatrix} \begin{pmatrix} X \\ Y \\ Z \end{pmatrix} = \mathbf{KP_c} = \mathbf{KT_{cw}P_w} \tag{5}$$

where $K$ is the internal parameter of the camera. We first use LiDAR mapping algorithm [9] to construct the complete point cloud map $\mathbf{P_w}$, and then use GT pose $\mathbf{T_{cw}}$ to transform the point cloud map into the camera coordinate system to obtain $\mathbf{P_c}$.

Since the event camera has the same optical lens principle as the traditional camera, we can obtain the point cloud within the viewing frustum of the event camera through the camera projection model (Eq. 5). However, camera projection for all point clouds is computationally expensive. Equivalently, we construct a camera model in the world coordinate system and use this camera model to construct a quadrangular viewing frustum. By applying four sets of plane equations, we can filter the range of point cloud coordinates effectively. Each screening operation significantly reduces the number of point clouds that require future calculations. Experimental results show that the calculation time is reduced by more than half when selecting points in the quadrilateral viewing frustum compared with projecting all points and performing visibility filtering. The detailed experimental setup in two different datasets is as follows:

**MVSEC** [7] is a well-known event-based dataset. MVSEC provide data on various motion mode (carried on a handheld rig, flown by a hexacopter, driven on top of a car and mounted on a motorcycle) in various scenarios. Since only indoor scenes provide high-precision pose and high-quality point cloud data, we only use indoor scene data for E2P tasks. We use the indoor-$x$ and indoor-$y$ sequences for training and testing respectively, where $x \in [1, 3]$ and $y = 4$. This experimental setup would generate a total number of 20,400 event (image) and point cloud pairs with the corresponding pose for training and 1,610 pairs for testing. In addition, each point cloud is augmented by adding random rotation (up to 0.5 degrees) and translation (up to 0.01 m).

**VECtor** [8] is the first event-based SLAM benchmark dataset captured by a full hardware-synchronized sensor suite. We use the VECtor dataset mainly to test the E2P effect in large-scale indoor scenarios. The VECtor dataset contains three different scenes of campus building interiors, each captured by two different motion modalities. We use the units-dolly, units-scooter, corridors-dolly and corridors-walk sequences for training, and the school-dolly and school-scooter sequences for evaluation. This experimental setup would generate 3,025 pairs for training and 1,544 pairs for testing. It is worth noting that our experimental setup uses completely different scenarios when training and testing, which will verify the generalization of our EP2T method to new scenarios. The same data augmentation method as MVSEC is also used.

### A.1.4 Datasets Preprocessing for Generalization Experiments

**MVSEC** [7] is also evaluated for optical flow estimation. However, the optical flow GT is sparse due to using the pose and depth map provided by LiDAR. Follow [2, 3], we use the $outdoor\text{-}day\text{-}2$ sequence (more than 12K grayscale frames) for training and test it on the $outdoor\text{-}day\text{-}1$ sequences. During training, we randomly select a processing window that spans 1/45 second, from which we extract all events and the optical flow GT. However, the training data will be discarded if the number of events in the window less than 10,000. It should be noted that the starting position of the processing window is chosen randomly, so it may not be aligned with the timestamp of the optical flow GT. To address this special condition, an interpolation operation may be necessary. For event-to-image reconstruction, we use the same sequence as flow estimation for training and testing.

**ECD** [10] contains a set of the asynchronous event stream, intensity images at about 24Hz, GT camera poses from a motion-capture system with sub-millimeter precision at 200Hz. As introduced in BTEB [11], we use the same sequences cut for testing. Unlike the MVSEC dataset, since BTEB does not require GT reconstructed images as a supervisory signal, we randomly but continuously sample the entire event sequence (N = 8,192) during training. During testing, we extract all events between image frames, and after random sampling, we use these sparse event data to reconstruct an image.

**N-Caltech101 and N-CARS** [12, 13] are two large public datasets for event-based classification. N-Caltech101 is an event camera version of the Caltech101 [12] dataset, which was created by moving an event camera that focused on an LCD monitor displaying the original Caltech101 data. Following the example of the original paper [12], we randomly select 15 samples from each class for testing. In N-CARS dataset, an event camera was mounted behind the windshield of a moving car during collection, and each sample contains 59,249 events. The whole dataset comprises 12,336 car samples and 11,693 background samples, which divided 7,939 cars and 7,482 backgrounds for training and others for testing. Similarly, we sample 8,192 events from a large number of events in each sample.

## A.2 Evaluation Metrics

In order to evaluate the generalization of our proposed EP2T module, we employ multiple evaluation metrics [2, 14, 15]to compare the results of different tasks.

**Flow Estimation.** Average end-point error (AEE [2]) denotes the distance between the end-points of the predicted $(Y')$ and the GT $(Y)$ flow vectors:

$$AEE = \sum_{x,y} \left\| \left( \begin{array}{c} u(x,y)_{Y'} \\ v(x,y)_{Y'} \end{array} \right) - \left( \begin{array}{c} u(x,y)_{Y} \\ v(x,y)_{Y} \end{array} \right) \right\|_2 \tag{6}$$

where $u, v$ represent the horizontal and vertical optical flow value respectively. Follow [2, 3], we limit the computation of AEE to pixels in which at least one event was observed. In addition to pixel-level evaluation, we also perform a global outlier ratio analysis. Our experiments regard pixels with $AEE > 3$ as outliers.

$$Outlier = \frac{\sum_{i=1}^{H \times W} \mathbb{1}\left(AEE_i > 3\right)}{H \times W} \times 100\% \tag{7}$$

where $H$ and $W$ denotes the solution of the flow image, $\mathbb{1}(\cdot)$ is the indicator function, which equals 1 if the AEE of pixel i is greater than 3, and 0 otherwise.

**Event-to-Image Reconstruction.** Mean squared error (MSE) is a classic evaluation metric to compare the distance of two different images. Unlike AEE (defined in Eq. 6), the reconstruction task is to evaluate all pixels of the image (regardless of whether event exists).

$$MSE = \frac{1}{n} \sum_{i=1}^{H \times W} \left\| \vec{Y_i} - \vec{Y_i'} \right\|_2 \tag{8}$$

where $Y$ is the GT grayscale image and $Y'$ is the reconstructed image by event-based methods. Similarly, in addition to pixel-level evaluation, reconstruction tasks often focus on the semantic

Table 1: **Registration performance (DeepI2P as baseline in MVSEC-E2P dataset) under different sampling methods.**

| Method | Uniform | Random | Farthest Point Sampling [17] | Voxel | Surface Event Sampling [18] |
|---|---|---|---|---|---|
| RE($°$)($\downarrow$) | 5.127 | 12.215 | 6.521 | 8.622 | **5.095** |
| TE(m)($\downarrow$) | **0.164** | 4.200 | 0.249 | 0.212 | 0.166 |
| Time(ms)($\downarrow$) | 1.1 | **0.3** | 796 | 8 | 421 |

similarity (LPIPS [15]) and structural similarity (SSIM [14]) between the reconstructed image and the GT image to prevent perceptual differences. LPIPS is an image quality evaluation index based on deep learning, which measures the semantic similarity between two images, defined as follows:

$$\text{LPIPS} = \frac{1}{N} \sum_{i=1}^{N} \text{D}_{\text{net}} \left( Y_i', Y_i \right) \tag{9}$$

where $D_net$ represents the distance or difference metric between image patches computed by a pre-trained deep network (AlexNet [16] in experiments) , N is the number of patch while the patch size is $32 \times 32$. SSIM measures how similar two images are in terms of structure, brightness, and contrast, detailed defined below:

$$\text{SSIM} = \frac{\left(2\mu_{Y'}\mu_Y + C_1\right)\left(2\sigma_{Y'Y} + C_2\right)}{\left(\mu_{Y'}^2 + \mu_Y^2 + C_1\right)\left(\sigma_{Y'}^2 + \sigma_Y^2 + C_2\right)}$$
$$C1, C2 = (k_1 L)^2, (k_2 L)^2 \tag{10}$$

where $\mu$ denotes the mean of pixel values, $\sigma$ denotes the standard deviation of pixel values, and $\sigma_{Y'Y}$ indicates the covariance between pixel values of two images. C1 and C2 are constants for stable calculations, L is the dynamic range of pixel values (255 in common) and $k_1 = 0.01, k_2 = 0.03$.

**Object Recognition.** Accuracy is a commonly used evaluation metric for classification model performance, which is used to measure the proportion of samples correctly classified by the classification model in the prediction process. The simple definition is as follows:

$$\text{Accuracy } = \frac{\text{M}}{\text{N}} \times 100\% \tag{11}$$

where $M$ indicates the number of samples correctly classified by the classification model in the prediction, $N$ represents the total number of predicted samples.

### A.3  Additional Results

**The impacts of sampling methods on performance.** To better demonstrate the performance of E2P under different aggregation centers, we further report the results under different sampling methods. As shown in Tab. 1, we compare the performance between Uniform Sampling (US), Random Sampling (RS), Farthest Point Sampling (FPS [17]), Voxel Sampling (VS) and Surface Event Sampling (SES [18]). For a fair comparison, we choose DeepI2P as the baseline and evaluate in MVSEC-E2P dataset. Experiments show that the simple uniform sampling method used in EP2T is relatively excellent regarding time efficiency, only slightly slower than the spatio-temporal random sampling method. In addition, because uniform sampling overcomes the inhomogeneity of local redundancy and global sparseness of event data, the registration precision also shows good potential, which is comparable to the learning-based event surface sampling method.

**Qualitative examples.** Here, we provide visualizations in Fig. 1 and Fig. 2 to present the effect of our EP2T module under different vision-based tasks (optical flow estimation, event-to-image reconstruction). Specifically, we demonstrate the event-to-image reconstruction effect of the EP2T module on the ECD [10] dataset with BTEB [11] as the baseline. As the limitation we described, it would be unwise to take all events between image frames for reconstruction like most frameworks but randomly sample 8,192 events. From qualitative experiments, it can be seen that even if the sampling operation reduces global information, our framework still maintains good reconstruction performance (see Fig. 1). Similarly, we also demonstrated the optical flow estimation effect of the cGAN [19] network as a baseline under the MVSEC [7] dataset. Dense optical flow denotes flow with all pixels, while sparse optical flow denotes masked flow at the pixels with events (see Fig. 2).

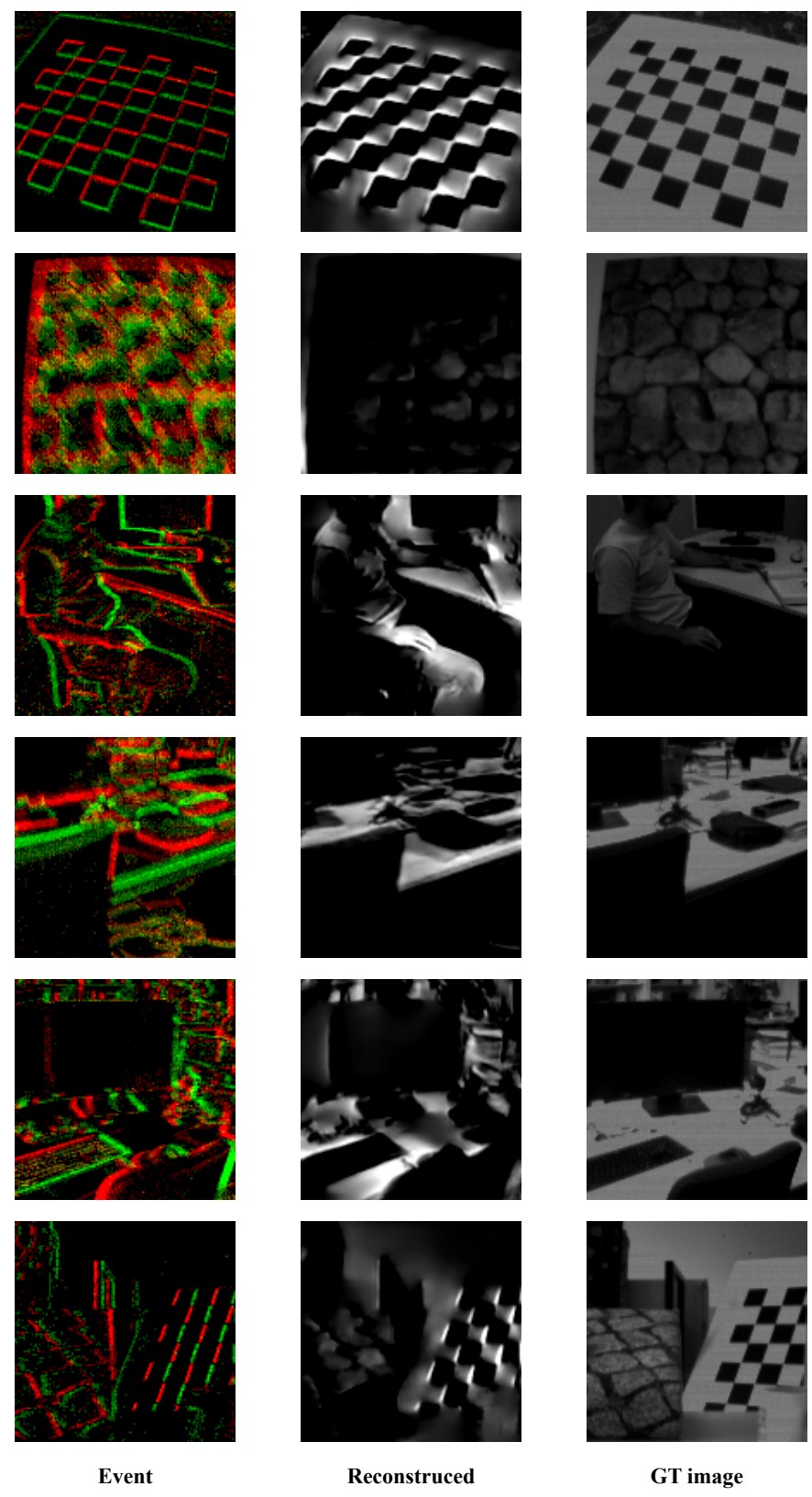

Event           Reconstruced           GT image

Figure 1: **Example of event-to-image reconstruction.**

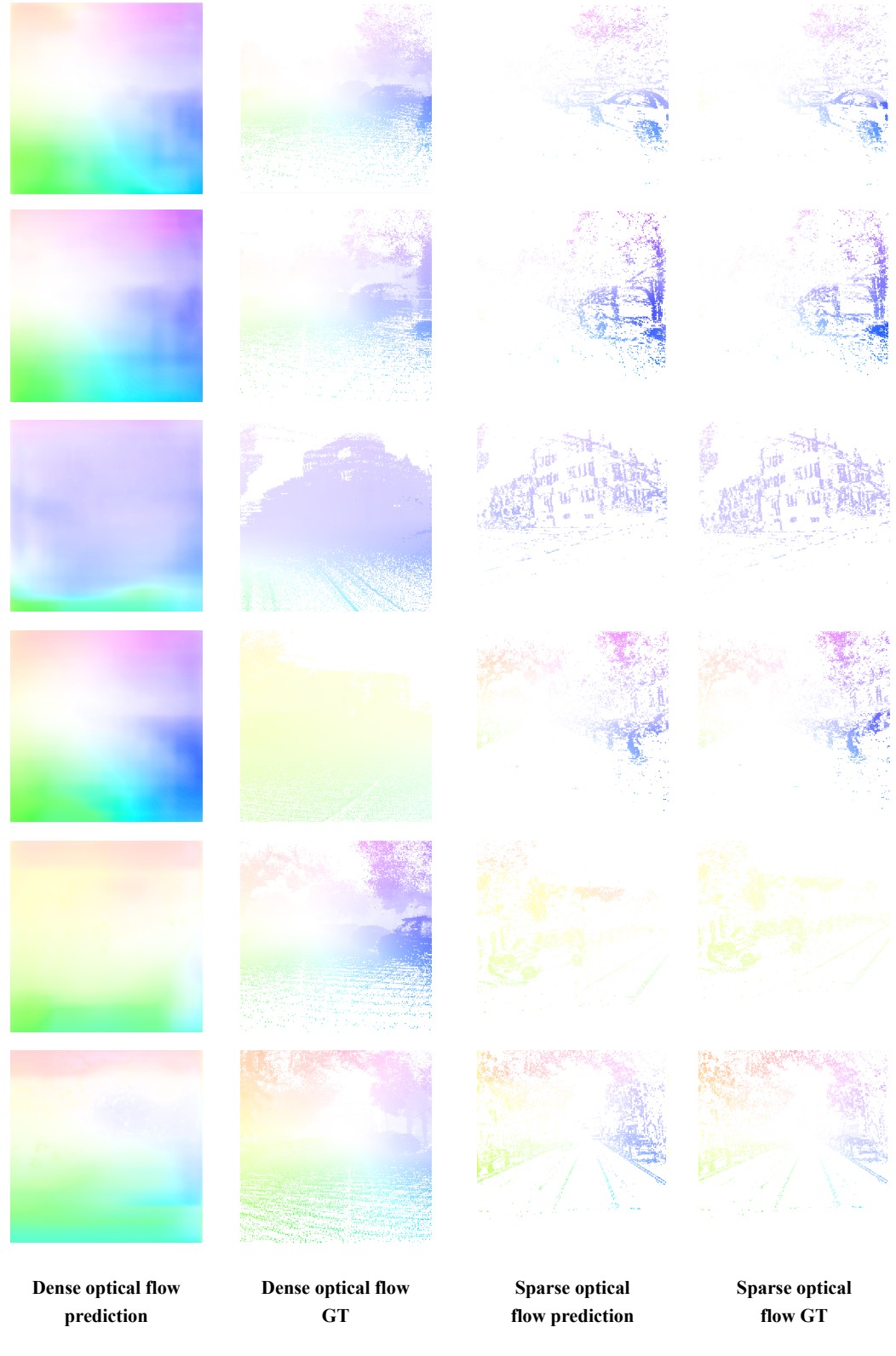

| **Dense optical flow prediction** | **Dense optical flow GT** | **Sparse optical flow prediction** | **Sparse optical flow GT** |

Figure 2: **Example of optical flow estimation.**