# OpenReview forum: "E2PNet: Event to Point Cloud Registration with Spatio-Temporal Representation Learning"
_NeurIPS.cc/2023/Conference — NeurIPS 2023 poster_

### Official Review · Reviewer_mSRE · 2023-06-29

**Soundness:** 2 fair
**Presentation:** 1 poor
**Contribution:** 3 good
**Rating:** 7
**Confidence:** 3

**Summary:**

The paper presents an method to perform point cloud registration using event camera data.
The method first learns a feature representation (E2PT) from a point cloud of events. This representation is used as input to standard registration networks.
The experimental section shows better accuracy compared to other event-based feature represenations as well as frame-based approaches.
Other experiments also show that the E2PT representations is generic and cna be used for other event vision tasks.

**Strengths:**

The paper addresses a novel problem of using event camera data for 2d-3d registration. They adapt existing registration pipelines taking into account the specificity of the events.
They propose a new learned event-based representation which leads to good experimental results for the registration problem, but also for other event vision tasks (optical flow, classification, image reconstruction).

**Weaknesses:**

The clarity of the paper can be overall improved.
In particular the method section is not always clear and some details are missing.
For example it is not clear how the features Ft , Fs , and Fst are built (Sec. 3.1.1). the authors mention some "multi-layer convolution", but is not clear what is the input to these convolutions, what exactly is the architecture, if tehre is a separate network for  Ft , Fs , and Fst or if it is shared. What is the size of Ft , Fs , and Fst etc. Describing the algorithm with more equations and pseudo-code can make the method clearer, since text and figures alone can be ambigous.

Similarly there is no description in the method section of the E2Pnet architecture, which is presented as the second main contribution in the introduction. From the experimental details it seems that standard architecture from previous works [1,11] are used?

**Questions:**

Please describe more precisely the steps of the method as stated above.

Also, please clarify the event sampling strategy: "We follow the FEN [25, 26] principle and acquire 20000 consecutive
events at a time and sample N = 8192 events from them."
Does it mean that out of 20k events, only N are used to build the tensor representation or that multiple consecutive tensor representations are used? why using such a specific value for N: 8192?

**Limitations:**

It would be interesting to understand the runtime of this approach, and how does it scale with respect to the event rate (number of events in the point cloud).

---

> ### Author Rebuttal · Authors · 2023-08-09
>
> Thanks for the thorough review and valuable suggestions. Due to space constraints, detailed explanations of the methodology are contained in Sec. A.1.1 of the appendix, including the technical details about the LA, STA, and FP modules. Our approach builds upon the 3D point-based architecture (PointNet++ [1]) and has been designed to leverage the unique characteristics of the event spatio-temporal data. For example, we introduce the spatio-temporal separation mechanism, which is explained in Sec. 3.1 of the paper. Below, we provide further details to better address your concern. We will carefully modify the camera-ready paper to make the method description more straightforward.
>
> **Weaknesses (How are the features $F_{t}$, $F_{s}$, and $F_{st}$ built? What exactly is the architecture? )**
>
>  $F_{t}$, $F_{s}$, and $F_{st}$ are obtained by the LA module with different parameters. The LA module needs to find the corresponding K nearest neighbors for each feature aggregation center (sampling point) to perform local feature aggregation (LA). Traditional 3D point cloud approaches usually use Euclidean distance to determine nearest neighbors. In this paper, we consider the different physical meaning and distribution characteristics of event data in time and space. Hence, we use different weight parameters for time and space, and the neighborhood of each sampling point is determined according to the weighted distance. The features generated by LA with larger temporal and spatial weights are called $F_{t}$ and $F_{s}$, respectively. The features generated by LA with the same temporal and spatial weights are called $F_{st}$. Please refer to lines 180 to 190 of the main paper for a more details.
>
> The structure of LA is similar to PointNet++ [1]. For each set of spatio-temporal distance weights, after determining the K nearest neighbors of each sampling point through the weighted spatio-temporal distance, we can obtain a feature matrix of size $S\*K\*3$; $S$ represents the number of aggregation centers, $K$ is the number of neighborhood points (64 in our experiments) and $3$ corresponds to the space-time coordinates ($h, w, t$) of the event.
> The features are mapped to a high-dimensional space through a 3-layer MLP to obtain features of size $S\*K\*C$, where $C=3,16,32,64$ represents the number of feature channels. Each layer of the MLP is followed by a layer of batch normalization and ReLU activation function. After that, we perform maximum pooling on the $K$ dimension, retain the neighborhood point with the largest feature value in each feature channel, and finally obtain the $S\*64$ dimensional feature.
>
> **Weaknesses (No description in the method section of the E2PNet architecture, standard architecture from previous works used?)**
>
> The main focus of this work is the feature representation learning of the event data. While E2PNet can utilize different mature 2D-3D registration frameworks after obtaining the output from EP2T (line 152 of the paper), we do not propose a novel registration framework. In our experiments we tested the classical 2D-3D registration frameworks DeepI2P[2] and LCD[3]. We will make this more clear in the camera-ready paper and merge the contribution statement of E2PNet with EP2T to avoid ambiguity.
>
> **Questions (Does it mean that out of 20k events, only N are used to build the tensor representation or that multiple consecutive tensor representations are used? Why use such a specific value for N: 8192?)**
>
> As correctly pointed out, we sample N=8192 events from the original consecutive 20k events. Empirically, this strikes a good balance between efficiency and high accuracy.
> To further ablate our choice, we have conducted an analysis of the impact of these two hyperparameters (number of FEN and number of sampling points) from three perspectives: accuracy, time efficiency, and memory cost. The results can be found in the newly uploaded rebuttal PDF. Overall, the proposed hyperparameter settings makes E2PNet reasonably efficient (time and memory) yet significantly more accurate than the baselines. In practice, the value of N can be increased (see the result of 16384 in Tab. 1 of the rebuttal PDF) to achieve higher accuracy or decreased (512 in Tab. 1 of the rebuttal PDF) to achieve much higher efficiency at the cost of accuracy (still better than the baseline).
>
> **Limitations (Runtime of this approach and how it scales with respect to the event rate)**
>
> According to your suggestion, we conducted further analysis of runtime in the rebuttal PDF. Experiments demonstrate that our method achieves superior accuracy and acceptable overhead in terms of runtime and memory consumption. In terms of the scaling, the runtime and memory cost increase with the number of sampling points, with the accuracy also consistently improving.
>
> We also identify some recently proposed techniques to optimize the efficiency of PointNet++, such as delayed aggregation [4] (2.2x speed up on PointNet++, using ModelNet40 [5] dataset), Sparse convolution [6] (20x acceleration on VGG13 [7], using N-Cars [8] dataset). Since our EP2T is inspired by PointNet++, a similar improvement can be expected by applying the same techniques, which will be an interesting direction for future work.
>
> >[1] Qi, et al. "Pointnet++: Deep hierarchical feature learning on point sets in a metric space."
>
> >[2] Li, et al. "DeepI2P: Image-to-point cloud registration via deep classification."
>
> >[3] Pham, et al. "Lcd: Learned cross-domain descriptors for 2d-3d matching."
>
> >[4] Feng, et al. "Mesorasi: Architecture support for point cloud analytics via delayed-aggregation."
>
> >[5] Wu, Zhirong, et al. "3d shapenets: A deep representation for volumetric shapes."
>
> >[6] Messikommer, et al. "Event-based asynchronous sparse convolutional networks."
>
> >[7] Simonyan, et al. "Very deep convolutional networks for large-scale image recognition."
>
> >[8] Sironi, et al. "HATS: Histograms of averaged time surfaces for robust event-based object classification."

---

> > ### Comment · Reviewer_mSRE · 2023-08-16
> >
> > Thank you for your answers. I have no further questions for now

---

> > > ### Author Response · Authors · 2023-08-19
> > >
> > > Thanks for your response, we are happy to be able to address your questions!

---

### Official Review · Reviewer_N3N6 · 2023-07-04

**Soundness:** 2 fair
**Presentation:** 2 fair
**Contribution:** 2 fair
**Rating:** 5
**Confidence:** 4

**Summary:**

This paper proposes a learning-based event-to-point cloud registration method, which encodes event spatio-temporal data into a grid-shaped feature tensor, and propose a framework to construct E2P datasets using existing SLAM datasets. Experiments are conducted on MVSEC-E2P and VECtor-E2P datasets, and state-of-the-art results are achieved on these datasets.

**Strengths:**

1. The presentation is easy to understand.
2. Experiments are well conducted and convincing.

**Weaknesses:**

1. Since event-based applications require a high response speed, the efficiency analysis should be given about the comparison of the runtime and memory usage between the proposed method and other state-of-the-art methods.
2. Since the experimental datasets are generated by the authors, did the authors retrain the learning-based models on the datasets in order to compare with these learning-based methods?
3. The comparisons are done to a few methods in the current manuscript, I suggest more state-of-the-art methods should be included for comparison.
4. Minor Typos / Writing
Figure 3: what does O_{E} mean?
Line 211: the definition of f_{SP} is missing.

**Questions:**

See Weaknesses.

**Limitations:**

Yes

---

> ### Author Rebuttal · Authors · 2023-08-05
>
> Thanks for the positive comments about our writing and experiments. In the following, we address the reviewer's concerns and back up our responses with additional experiments. We hope that with major concerns like efficiency analysis and training details resolved, the reviewer will consider improving the final ratings.
>
> **Comparison of the proposed method and other state-of-the-art methods w.r.t. runtime and memory usage.**
> Thanks for the suggestion, we have uploaded a new PDF analyzing the time and space efficiency of E2PNet. Though slightly slower (110ms *VS.* 55ms inference time) and more memory-intensive (7.98GB *VS.* 2.2GB) compared to direct tensor-based approaches, given the significantly improved accuracy, this overhead is arguably acceptable for the registration task.
>
> Note that the efficiency of the introduced E2PT network can be further improved by incorporating recent techniques, e.g., 2.2x potential speedup from the delayed aggregation [1]. We are committed to investigating this interesting direction to improve our method in the future. We also show in Tab. 1 of the rebuttal PDF that one can reduce the number of center points in EP2T to significantly reduce the speed and memory consumption (512 points result in 65ms inference speed and 2.9GB memory usage) while still achieving better performance than baselines.
>
> We will add this discussion to the camera-ready paper.
>
> **Did the authors retrain the learning-based models?**
> Yes. To fairly compare against different methods, we have retrained all learning-based baselines on the proposed datasets. We will clarify this in the camera-ready paper.
>
> **The comparisons are done to a few methods in the current manuscript. I suggest more state-of-the-art methods should be included for comparison.**
>
> Thanks for the suggestion. This work focuses on the feature representation learning of event-to-point cloud registration. To demonstrate the effectiveness of our E2PT representation, we have compared it against hand-crafted features (discretized event volume [2], published at CVPR2019) and advanced methods with both tensor-based representations (Tore [3], published at T-PAMI 2023) and point-based representations (ECSNet [4], published at T-CSVT 2022). All of these methods are state-of-the-art in their respective domains and serve as solid benchmarks for evaluating the performance of our proposed event representation.
>
> To verify the effectiveness of our method under different 2D-3D registration frameworks (used after the feature representation network), we also compared against different representations under representative methods based on both registration (DeepI2P [5], published at CVPR2021) and retrieval (LCD [6], published at AAAI2020). Though there are other potential candidate registration frameworks like P2Net [7] (ICCV2021) and Pump [8] (CVPR2022), they did not release the code and our main focus is designing a better feature representation learning network that can be plugged into different downstream registration frameworks.
>
> **Typo/writing** We appreciate your attention to the details of our work. In Fig. 3, $O_{E}$  refers to the attention feature obtained from the spatio-temporal feature $F_{st}$ at each feature aggregation center through a classic self-attention module ($O_{E}$ has the same dimension as $F_{st}$ ). Our approach involves not only assigning local neighborhood features to each aggregation center through the LA module but also considering the importance of global information. To achieve this, we introduce a self-attention module to enhance the long-distance correlation between the spatio-temporal features of each aggregation center, which allows for better information interaction among all centers.
>
> The $f_{SP} $ in the paper is a typo, and it should be $f_{FP}$, which represents the point-wise spatio-temporal features obtained after feature propagation (FP module).
>
> We will fix/clarify these points in the camera-ready paper and carefully proofread the manuscript once more.
>
> >[1] Feng, et al. "Mesorasi: Architecture support for point cloud analytics via delayed-aggregation."
>
> >[2] Zhu, et al. "Unsupervised event-based learning of optical flow, depth, and egomotion."
>
> >[3] Baldwin, et al. "Time-ordered recent event (TORE) volumes for event cameras."
>
> >[4] Chen, et al. "ECSNet: Spatio-Temporal Feature Learning for Event Camera."
>
> >[5] Li, et al. "DeepI2P: Image-to-point cloud registration via deep classification."
>
> >[6] Pham, et al. "Lcd: Learned cross-domain descriptors for 2d-3d matching."
>
> >[7] Wang, et al. "P2-net: Joint description and detection of local features for pixel and point matching."
>
> >[8] Revaud, et al. "Pump: Pyramidal and uniqueness matching priors for unsupervised learning of local descriptors."

---

> > ### Comment · Reviewer_N3N6 · 2023-08-19
> >
> > Thank you for the author's feedback. Based on Table 2 in the Author Response Appendix, instead, I think the increases in runtime and memory consumption of the proposed method do not result in significant performance gains.

---

> > > ### Author Response · Authors · 2023-08-19
> > > **The significance of performance gain**
> > >
> > > Thanks for the response. In terms of the performance gain, we kindly argue that **the performance gain is significant**.
> > >
> > > 1) In Tab. 2 of the response appendix, E2PNet has reduced the average error of LCD (Grayscale Image) relatively by **39%** (1.35m to 0.82m) for translation and **43%** (6.336 degree to 3.606 degree) for rotation. Comparing to LCD + other features, E2PNet also **at least** provides a relative error reduction by **35%** for translation (vs 1.263m ECSNet) and **27%** (vs 4.968 degree of E-statistic) for rotation. All of these improvements are non-trivial and significant. Note that the performance difference between other baselines are **much smaller** than the improvement that E2PNet provides.
> > >
> > > 2) In Tab. 1 of the response appendix, we also show that E2PNet can be tuned to trade accuracy with efficiency. Specifically, by setting EP2T sampling points to 512 (rather than 8192 by default), we have a **similar runtime and memory consumption** as the baselines, yet still being significantly more accurate than **all** baselines. E.g., comparing to LCD (Grayscale Image), we still have **40%** relative reduction for rotation error and **19%** translation error improvement. This shows the effectiveness of E2PNet under various speed and memory constraints.
> > >
> > > 3) E2PNet also has **a much lower registration failure rate** comparing to the baselines. For example, on the same dataset as in the rebuttal appendix, we compute the ratio of scenes that LCD (Grayscale Image) and E2PNet can register with translation error of <1m and rotation error of <1 degree. The success rate of E2PNet is 59% yet LCD only has 32%, which is **nearly half of the success rate of E2PNet**. We will add this result into the camera ready.
> > >
> > > With all these results, we kindly argue that **the performance improvement of E2PNet is significant, and the accuracy improvement does not necessarily comes with overheads**.
> > >
> > > Considering this result and the fact that we have addressed other questions/concerns, we kindly ask the reviewer to re-consider the ratings. We are happy to address further concerns/questions if the reviewer has any.

---

### Official Review · Reviewer_W8Di · 2023-07-06

**Soundness:** 4 excellent
**Presentation:** 3 good
**Contribution:** 4 excellent
**Rating:** 6
**Confidence:** 4

**Summary:**

This paper proposed a Event-Points-to-Tensor (EP2T) network, which treats event data as spatio-temporal point clouds, to process event signals without losing the spatiotemporal information of event signals (especially temporal information, compared with other voxel grid-based methods). In terms of experiments, this work demonstrates the effectiveness of the EP2T network by using event-based point cloud registration as an example, resulting in the development of E2PNet. Furthermore, the authors have gone a step further and tested the generalization ability of the EP2T network in tasks such as optical flow estimation, image reconstruction, and object recognition, obtaining promising results.

**Strengths:**

The proposed Event-Points-to-Tensor (EP2T) network in this paper takes a different approach compared to most existing event signal processing models. This method treats event signals as three-dimensional spatio-temporal point clouds and employs operations such as Local Aggregation (LA), Spatio-temporal Separated Attention (STA), and Feature Propagation (FP) for extracting and preprocessing spatio-temporal features of events. Finally, tensorization is performed to obtain event feature representations compatible with traditional visual models. Compared to previous models that always tensorize event signals before feeding them into the model, this method can fully utilize the discrete and sparse characteristics of events. It effectively extracts spatiotemporal features from events, especially rich temporal domain information. Additionally, the LA operation separates the extraction of features in the temporal and spatial domains, enabling feature extraction and aggregation on different dimensions/domains, considering the distinct physical meanings of the three dimensions in the event signal "point cloud".

**Weaknesses:**

Although the proposed EP2T network in this paper achieves sparse processing of event signals in the first half, it still requires tensorization of the events in the end. In practice, such an operation can significantly degrade the temporal precision of events. While the 2D-3D Registration task proposed in this paper may not aim for high temporal resolution, it is crucial for tasks that are sensitive to event precision, such as high-frame-rate video reconstruction and low-latency object tracking. Furthermore, although the idea of using point cloud networks for sparse processing of event signals in this paper is innovative, the intuitive motivation behind this approach is not clearly presented in the writing. For instance, compared to the operation of tensorization followed by feature extraction, it is not discussed in the paper which events EP2T can handle better and what advantages EP2T possesses. This discussion is lacking in the paper.

**Questions:**

The authors are encouraged to respond to the discussion regarding the intuitive motivation behind EP2T and provide an explanation of whether this operation takes into account the mathematical/physical characteristics of events better or offers better feature extraction advantages compared to traditional dense CNN-based model structures, or if there are other considerations involved. Furthermore, the reviewers request additional targeted experiments to validate the proposed intuitive advantages.

**Limitations:**

The author points out that the problem of large memory consumption is an important future research direction, which is similar limitation to other point-based models. If this problem can be solved, it will indeed help to improve the practicability of the model. Furthermore, there is a potential limitation that the authors did not mention in the paper, namely, that E2PNet still relies on tensorization to convert event point clouds into dense grid representations, which are then fed into downstream models designed for traditional vision tasks. It would be intriguing if future work could bypass this step and directly output target results using discrete/sparse event signals.

---

> ### Author Rebuttal · Authors · 2023-08-05
>
> We thank the reviewer for recognizing the value of this work and providing an in-depth review. We provide the responses to the questions/concerns below.
>
> **The motivation, advantage, and limitation of EP2T+tensorization-based approaches** First of all, there is no existing event to point cloud registration framework. However, several 2D-3D (tensor-based, e.g., LCD [1] and DeepI2P [2] in our experiments) and 3D-3D (point-based, e.g., GeoTransformer [3]) frameworks exist for other modalities. Due to the direct use of spatial geometry to perform registration, 3D-3D frameworks are not easily applicable to event data since the event signal does not have 3D spatial information. 2D-3D registration frameworks can be directly applied to event data but most temporal information is lost. Our key idea is to design a feature representation network that can effectively extract the spatio-temporal feature before applied to different registration frameworks, whether tensor-based or point-based. In Tab. 1 of the main paper, we have compared the performance of E2PT when applied to both tensor and point-based methods. As described in line 260 of the main paper, we have modified LCD [1] and DeepI2P [2] so that we directly convert point-based features into a global feature without using any grid-based tensorization. We agree that designing a better fully point-based registration framework for event data is an interesting research direction. However, we do not see an obvious advantage of pure point-based methods over EP2T+tensor-based methods. Hence, based on the empirical accuracy, we choose to construct our E2PNet using E2PT+tensor-based methods. We will state this motivation more clearly in the camera-ready paper.
>
> In terms of the advantage, focusing on the spatio-temporal feature learning makes it possible for our method to utilize different future registration frameworks, which can come from outside the field of event data processing. Since tensor-based methods are efficient, combining EP2T with tensor-based methods also provides reasonable speed and memory efficiency along with high registration accuracy.
>
> **Limitations on memory consumption** We appreciate your feedback on the limitation of memory consumption. To further analyze and address this limitation, we have uploaded a PDF that includes a detailed analysis of memory usage. As shown in Tab. 2 of the PDF, though E2PNet exhibits slightly lower speed (110.6ms *VS.* 55ms) and larger memory cost (7.89GB *VS.* 2.2GB), this overhead is arguably acceptable compared to the significantly improved accuracy (translation error decreased from 1.297m to 0.821m, rotation error decreased from 4.97 ° to 3.61°).
>
> Reducing memory consumption is indeed an important direction for future work. The major contributor to the higher memory usage is the EP2T network designed based on PointNet++ [4], where the point-wise distance calculation operation leads to memory growth that scales with the number of points, resulting in a quadratic consumption increase. There are successful approaches that can significantly improve the space efficiency of point cloud frameworks, such as PAT [5] (2x memory reduction on PointNet++, using ModelNet40 [6] dataset) and PVCNN [7] (3x memory reduction on point-based models, using ModelNet40 dataset). Most of these methods can be readily integrated into EP2T, offering the potential to enhance (memory) efficiency without compromising performance. Meanwhile, for applications that require high inference speed, we can reduce the number of EP2T sampling points (Tab. 1 of the rebuttal PDF) to achieve a significant memory reduction (2.99GB with 512 sampling points) while still performing better than the baselines. We are dedicated to further optimize the memory consumption of EP2T, and your feedback has reinforced our commitment.
>
>
> >[1] Pham, et al. "Lcd: Learned cross-domain descriptors for 2d-3d matching."
>
> >[2] Li, et al. "DeepI2P: Image-to-point cloud registration via deep classification."
>
> >[3] Zheng, et al. "GeoTransformer: Fast and Robust Point Cloud Registration With Geometric Transformer."
>
> >[4] Qi, et al. "Pointnet++: Deep hierarchical feature learning on point sets in a metric space."
>
> >[5] Yang , et al. "Modeling point clouds with self-attention and gumbel subset sampling."
>
> >[6] Wu, et al. "3d shapenets: A deep representation for volumetric shapes."
>
> >[7] Liu, et al. "Point-voxel cnn for efficient 3d deep learning."

---

> ### Author Response · Authors · 2023-08-19
>
> Hi, we understand that you are very busy. We just want to kindly remind that the discussion phase will pass soon, and we want to make sure that we have addressed all your concerns. Please leave us a comment if you still have further questions/concerns, we will try our best to address them. If we cannot response later due to the discussion deadline, we will address them properly in the camera ready. Looking forward to hearing from you!

---

> ### Comment · Reviewer_W8Di · 2023-08-20
>
> The rebuttal has addressed my comments. I would like to keep my rating unchanged.

---

### Official Review · Reviewer_mNvD · 2023-07-08

**Soundness:** 2 fair
**Presentation:** 3 good
**Contribution:** 2 fair
**Rating:** 5
**Confidence:** 5

**Summary:**

In this paper, the authors proposed the first learning-based work that can handle event-to-point cloud registration (E2P). More specifically, a novel Event-Points-to-Tensor (EP2T) network is proposed to encode the data from the event camera into features tensors in the form of a 2D grid. The temporal patch aggregation and spatial patch aggregation combined with spatio-temporal kernel are applied to obtain the global feature. Then based on the output gridded tensor, a standard 2d-3d feature based algorithm is applied to obtain the structure and motion of the final 3D reconstruction scene.

The experimental results based on the two representative datasets demonstrate the performance, mainly in terms of accuracy, of the proposed E2PNet on the task of event to point cloud registration.

**Strengths:**

The motivation of this paper is well designed, that is introducing the first learning-based architecture to handle the sparse reconstruction based on event cameras. More specifically, the spatial and temporal attention mechanism together with the feature propagation module to obtain the final output gridded tensor. The design of the EP2T network is straightforward, elegant and easy to follow.

In addition to its innovative approach, the paper stands out for its well-structured writing style that includes a clear and concise method statement. The authors have taken great care in presenting their research in a manner that is easily understandable to the readers. Furthermore, the inclusion of visual representations helps to further elucidate the concepts and techniques discussed in the paper.

Moreover, the performance shown in the statistical experiments especially in Table 1 demonstrate the superior performance of the proposed approach over the previous state-of-the-art approaches.

**Weaknesses:**

My major concerns lie in the following two aspects.

First, the statistical experiments in Table 1-3 only demonstrate the accuracy in terms of camera position and direction, while the efficiency especially the time and memory efficiency is missed.

Second, the limitations or failure case of the proposed approach need to be discussed. I am wondering the performance of the proposed approach on more complex and even comparatively large-scale environment.

**Questions:**

See the "Weakness" section.

**Limitations:**

See the "Weakness" section. Furthermore, the final score assigned to the study is not solely determined by the discussion among peer reviewers. If the authors can solve my main concerns, I would like to raise the score.

---

> ### Author Rebuttal · Authors · 2023-08-05
>
>
> Thanks for your positive comments about the novelty, writing, and experiments of this work. Please see the following responses to your concerns.
>
> **Time and memory efficiency** We have uploaded a new PDF containing an analysis of the time and memory efficiency of E2PNet. As shown in Tab. 2 of the new PDF, E2PNet is slightly slower (110.6ms *VS.* 55ms) and uses more memory (7.89GB *VS.* 2.2GB) than baseline methods. However, the overhead introduced by EP2T is arguably small given the significantly improved accuracy (translation error decreased from 1.297m to 0.821m, rotation error decreased from 4.97 ° to 3.61°). Note that further efficiency improvements are possible by 1) combining EP2T with more efficient tensorized representation methods, 2) tuning the number of spatial-temporal sampling points for efficiency-accuracy trade-off (Tab. 1 and 2 of the newly added PDF, show that when reducing the number of sampling points to 512, E2PNet runs at a similar speed (65.5ms) and memory cost (2.99GB) as other baselines, yet still has higher accuracy), and 3) optimization techniques like delayed aggregation [1] (2.2x speed up on PointNet++ [2], using ModelNet40 [3] dataset), Sparse convolution [4] (20x acceleration on VGG13 [5], using N-Cars [6] dataset). We will add this analysis and discussion into the camera ready following your suggestion.
>
> **Limitations or failure cases** As discussed in lines 325 to 328 of the paper and the previous response, the limitations of the proposed method lie mainly in the efficiency, which is a common problem in 3D point-based learning architectures. We will state this limitation more clearly by adding the efficiency analysis results to the camera-ready paper and provide further discussions as mentioned in the previous response.
>
> **Complex or Large-scale Environment** Exploring datasets with a more complex structure or a larger scale is indeed a significant problem. Since the registration task requires accurate ground-truth poses for supervision, which (at the moment) is only available in existing indoor data, we evaluate algorithms on indoor scenes in this work. However, we tried our best to involve large-scale and challenging scenes in the experiments (Tab. 1 of the main paper). The VECtor [7] dataset contains indoor scenes with corridors that have long edges of more than $60$m. We use different scenes in the training and testing data to better evaluate the generalization capability of different algorithms. We will discuss this limitation in the paper but leave the construction of better benchmarks for future work.
>
> >[1] Feng, et al. "Mesorasi: Architecture support for point cloud analytics via delayed-aggregation."
>
> >[2] Qi, et al. "Pointnet++: Deep hierarchical feature learning on point sets in a metric space."
>
> >[3] Wuc "3d shapenets: A deep representation for volumetric shapes."
>
> >[4] Messikommer, et al. "Event-based asynchronous sparse convolutional networks."
>
> >[5]  Simonyan, et al. "Very deep convolutional networks for large-scale image recognition."
>
> >[6] Sironi, et al. "HATS: Histograms of averaged time surfaces for robust event-based object classification."
>
> >[7] Gao L, et al. "Vector: A versatile event-centric benchmark for multi-sensor slam."

---

> ### Author Response · Authors · 2023-08-19
>
> Dear reviewer, thanks for your hard work. We understand that you might be very busy at the current moment. A kind reminder that the discussion phase will pass soon. We would like to ask whether our responses have addressed your previous questions/concerns? We are happy to have further discussions if you still have other comments. If we cannot response later due to the discussion deadline, we will address them properly in the camera ready. Looking forward to your reply!

---

### Author Rebuttal · Authors · 2023-08-09

We thank all reviewers for their positive comments about the novelty (R1, R2, R4), significance (R1, R2, R4), writing quality (R1, R2, R3), and experiments (R1, R2, R3, R4) of this work.

A common question/concern was the efficiency of the proposed method. We have conducted several experiments and provide a detailed analysis in the uploaded rebuttal PDF. The results show that our method is reasonably efficient while achieving a much higher accuracy compared to the baselines. We also identify and discuss different approaches to further improve the efficiency. Please refer to the individual responses for further details and also responses to other specific concerns. We appreciate all reviewers' feedback and will incorporate it in the final manuscript.

---

### Decision · Program_Chairs · 2023-09-21

**Decision:**

Accept (poster)

**Comment:**

This paper receives 4x positive ratings: 2x borderline accepts, 1x weak accept and 1x accept. The overall comments on this paper are positive. The reviewers think that this is the first learning-based architecture to handle the sparse reconstruction based on event cameras; The performance shown in the statistical experiments especially in Table 1 demonstrate the superior; This paper takes a different approach compared to most existing event signal processing models; It effectively extracts spatiotemporal features from events, especially rich temporal domain information; A new learned event-based representation which leads to good experimental results for the registration problem, but also for other event vision tasks (optical flow, classification, image reconstruction).